# Synthesis and Characterization of High Surface Area Transparent SiOC Aerogels from Hybrid Silicon Alkoxide: A Comparison between Ambient Pressure and Supercritical Drying

**DOI:** 10.3390/ma15041277

**Published:** 2022-02-09

**Authors:** Adane Muche Abebe, Gian Domenico Soraru, Ganesh Thothadri, Dinsefa Mensur Andoshe, Andrea Zambotti, Gulam Mohammed Sayeed Ahmed, Vineet Tirth, Ali Algahtani

**Affiliations:** 1Department of Materials Science and Engineering, School of Mechanical, Chemical and Material Engineering, Adama Science and Technology University, Adama P.O. Box 1888, Ethiopia; ganesh_reliez@yahoo.co.in (G.T.); dinsefa.mensur@astu.edu.et (D.M.A.); 2Department of Industrial Engineering, University of Trento, Via Sommarive 9, 38123 Trento, Italy; soraru@ing.unitn.it (G.D.S.); andrea.zambotti-1@unitn.it (A.Z.); 3Department of Mechanical Design and Manufacturing Engineering, School of Mechanical, Chemical and Material Engineering, Adama Science and Technology University, Adama P.O. Box 1888, Ethiopia; gmsayeed.ahmed@astu.edu.et; 4Mechanical Engineering Department, College of Engineering, King Khalid University, Abha 61421, Saudi Arabia; vtirth@kku.edu.sa (V.T.); alialgahtani@kku.edu.sa (A.A.); 5Research Center for Advanced Materials Science (RCAMS), King Khalid University, Guraiger, P.O. Box 9004, Abha 61413, Saudi Arabia

**Keywords:** aerogel, SiOC ceramics, sol-gel, supercritical CO_2_ drying and porous materials

## Abstract

In this article, highly porous and transparent silicon oxycarbide (SiOC) gels are synthesized from Bis(Triethoxysilyl) methane (BTEM). The gels are synthesized by the sol-gel technique followed by both ambient pressure and supercritical drying. Then, the portion of wet gels have been pyrolyzed in a hydrogen atmosphere at 800 and 1100 °C. The FT-IR spectroscopy analysis and nitrogen sorption results indicate the successful synthesis of Si-O-Si bonds and the formation of mesopores. From a hysteresis loop, the SiOC ceramics showed the H_1_ type characteristic with well-defined cylindrical pore channels for the aerogel and the H_2_ type for the ambigel samples, indicating that the pores are distorted due to the capillary stress. The produced gels are mesoporous materials having high surface areas with a maximum of 1140 m^2^/g and pore volume of 2.522 cm^3^/g obtained from BTEM aerogels. The pyrolysis of BTEM aerogels at 800 °C results in the production of a bulk and transparent sample with a slightly pale white color, while BTEM xerogels are totally transparent and colorless at the same temperature. At 1100 °C, all the aerogels become opaque brown, confirming the formation of free carbon and crystalline silicon.

## 1. Introduction

Aerogels have become one of the most interesting materials. They have outstanding properties, such as large specific surface area, high porosity, low density and thermal conductivity, which are attributed mostly to their mesoporous nanostructure [1,2]. These properties have made aerogels intriguing as thermal insulators, dielectrics for electronics [1,3,4], molecular sieves, catalysis supports, sensors and as energy storage media [5]. Kistler was the first to introduce aerogels in 1931 [6]. A basic aerogel synthesis involves three distinct steps: gelation, aging, and drying. Given the desired chemistry of the aerogels (e.g., SiO_2_, SiOC), the selected precursors are dispersed in a solvent and allowed to gel, thus forming a continuous network of solid particles throughout the liquid, called wet gel.

The most critical step in preparation of the aerogel is supercritical drying. It is a tricky process that eliminates the liquid that lies in the pores of the gel by taking it to the supercritical state, thus avoiding collapse of the solid network. Ideally, the microstructure of a dry aerogel is identical to that of the wet gel before drying. On the other hand, conventional drying is responsible for large deformations as capillary stresses arise at the liquid-vapour interface. This capillary pressure may reach as high as 100–200 MPa [5,7] due to the presence of micro- and mesopores (it is inversely proportional to pore radius), causing the wet gel to collapse, eventually cracking and changing the size and shape of the mesoporous network. In general, the main benefits of the supercritical drying over ambient pressure drying are (1) extremely porous monolithic crack-free aerogels are prepared and (2) the relatively very fast process. However, drying with supercritical CO_2_ has certain limitations, such as high processing costs for commercial applications, since drying is carried out at high pressures, low moisture stability and low mechanical resistance [8].

In general, three basic drying routes are usually utilized (shown in Figure 1); those are: (a) freeze-drying, where the solvent within the pores must pass the liquid–solid and solid–gas equilibrium curves; (b) evaporation or ambient pressure (it is the pressure of the airtight space in which a given wet gel is placed) drying, which involves crossing of the liquid–gas equilibrium curve; and (c) supercritical fluid drying, in which the system is brought into a supercritical state where there is no liquid-gas interface. The fluid is then slowly vented to the atmosphere. Since this last drying takes place in the absence of liquid–vapor interfaces, no capillary pressure is present. Therefore, the solid skeleton is maintained in a highly porous state after drying.

Silica aerogels are the most researched aerogel materials, which are often synthesized by the sol-gel processes using an alkoxysilane (typically tetraethoxysilane, TEOS) polymer precursor. The inclusion of carbon atoms into the silica network can provide improved mechanical, thermal, optical, and electrical properties [9,10] due to the formation of rigid carbidic carbon (CSi_4_) units with a denser atomic network than pure silica. This results in SiOC aerogels, which are synthesized from hybrid silicon alkoxides. Silicon oxycarbide networks contain Si, O, and C, sometimes named “black glasses”. The black color derives from excess C in the amorphous network [9,11]. Transparent SiOC glasses can be obtained when a dried gel is pyrolysized in the presence of hydrogen gas, where H_2_ reacts with the organic groups to form methane gas, thus removing the excess carbon [9,11]. Synthesis of highly porous SiOC glasses may have potential applications in dielectrics for integrated photonics [12], gas sensors [13,14], and materials for lithium-ion batteries [15], biosensors, and photovoltaic cells. Moreover, SiOC glasses have high temperature resistances up to 1200 °C either in oxidizing or in inert atmospheres [10].

This article deals with the synthesis of porous silicon oxycarbide aerogels and xerogels derived from a sol-gel procedure involving bis(triethoxysiyl)methane as SiOC precursor. Ceramization was achieved using H_2_ as pyrolysis gas to remove the excess carbon and produce transparent SiOC. Furthermore, the result of CO_2_ supercritical drying is compared to conventional drying to assess the effect of porosity and pore size distribution on the microstructural features of these materials.

## 2. Materials and Methods

### 2.1. Sol-Gel Process

Bis(triethoxysilyl)methane (C_2_H_5_O)_3_Si-CH_2_-Si(OC_2_H_5_) (BTEM) CAS 18418-72-9 was received from Fluorochem chemicals, Hadfield, England. Isopropanol alcohol (C_3_H_7_OH) CAS 6763-0 was received from J.T. Baker chemicals, Deventer, Netherlands; and ammonia (NH_3_) CAS 7664-41-7 was purchased from Deselab Line chemicals, Piombino Dese, Italy. BTEM (340.56 g/mol) and isopropanol (60.10 g/mol) chemicals are used as received without any further purification. The 12 M HCl and 16.5 M NH_3_ chemicals were diluted to 3 M and 13.4 M, respectively, for further use.

Considering silicon oxycarbide gels, the sol-gel process is based on a three reaction mechanism according to Figure 2. In hydrolysis, Si-alkoxide reacts with water to produce silanol groups and alcohol. During the following step, condensation reactions generate Si-O-Si bridges, water, and alcohol. These reactions take place in the presence of a solvent and catalysts. At the functional group level, the reaction of alkosilanes to form the silica network can be written as in Figure 2 [13,14,15,16,17,18]:

The first step to prepare a precursor gel is to choose the appropriate molar ratio of silicon alkoxide:solvent; silicon-alkoxide:catalyst(s); and hydrolysis-condensation reaction conditions such as time and temperature. Varying these processing parameters strongly affects the materials microstructure and the chemistry of the solid parts. In the synthesis of BTEM gels, the acid-base two-steps process was applied. In the present case, both acidic and basic solutions controlled the hydrolysis and condensation reactions. The molar ratios between Si-alkoxides/isopropanol/HCl(aq)/NH_3_(aq) were set to 1/5/8/0.5 [10,11].

The main synthesis procedure in producing a BTEM gel can be described in detail as follows. One mole of BTEM silicon alkoxide was dissolved in 4 moles of isopropanol solvent and then hydrolyzed with 8 moles of acidic water (3 mol/L HCl(aq)). After hydrolysis, condensation was induced by drop-wise addition of 0.5 moles of a solution 13.36 mol/L NH_3_(aq) and 1 mole of isopropanol. To activate the condensation reactions, 20% by mole of isopropanol alcohol was used to dilute the ammonia solution to control the reaction and to achieve a homogeneous system (shown in Figure 3). This method was adopted to control the condensation process by diluting the ammonia aqueous solution with isopropanol. The sols were then stirred for a period of time and then transferred to a plastic mold. The cover of the plastic mold was closed in order to avoid the evaporation of isopropanol, which is volatile at room temperature. The surfaces of the obtained gels were kept wet with the drop-wise addition of isopropanol. The gels, covered by a thin layer of isopropanol and sealed in plastic mold, were aged for one week at 50 °C in a controlled atmosphere before swapping the synthesis solvent with neat isopropanol. The purpose of washing at this stage is mainly to remove the excess water present in the catalyst solutions and the one produced during condensation reactions. Finally, drying was accomplished via supercritical drying and ambient pressure drying.

Capillary pressure is responsible for the shrinkage of the SiOC gels during ambient pressure drying, and it is proportional to the surface tension of the utilized solvent. Isopropanol was selected for the SiOC gel synthesis, as it has a reasonably lower surface tension at 50 °C compared to other solvents [16]. After three weeks of complete gelation, a small pinhole was made on the plastic mold cap for the isopropanol to slowly vent out.

Similarly, in supercritical drying, the isopropanol was used to wash out excess water exploiting the complete miscibility between the two liquids. Then, the wet gel was placed into a customized supercritical CO_2_ reactor, where the extraction of the isopropanol, its substitution with liquid CO_2_ and the final drying of the gel took place. The extraction through solvent exchange was performed two times per day for about a week. Once the amount of isopropanol reached a negligible amount, the temperature of the reactor was increased to 45 °C, which is above the critical point of carbon dioxide. As a matter of fact, at this condition there is no distinction between liquid and vapor CO_2_. Finally, after maintaining the whole system for about an hour at 45 °C (at a pressure of 100 bar), carbon dioxide was slowly vented out, and the dried-gel was obtained. Table 1 shows the different designations for ambient and supercritical dried gels.

### 2.2. Pyrolysis Process

The dried SiOC gels were placed in an alumina boat and pyrolyzed in H_2_ atmosphere up to 800 or 1100 °C for 1 h at the maximum temperature. This heat treatment had been done in Japan by Prof. M. Narisawa at the Osaka Prefecture University. Based on previous studies [11], the samples were heated at 5 °C/min until the temperature reached 600 °C, the sample was kept for 1 h and then heated at 3.33 °C/min up to the desired temperature. Finally, a 600 °C/h cooling rate was employed to bring the system to room temperature.

### 2.3. Characterization

The skeletal density of all the gels was calculated by he-Pycnometry using Micromeritics AccuPyc 1330 equipment (Micromeritics, Norcross, GA, USA).

Fourier transform infrared (FT-IR) spectra were recorded with a Varian Excalibur Series 4100 (Lake Forest, CA, USA) at room temperature using the ATR mode with a diamond crystal in the 4000–500 cm^−1^ range, acquiring 64 scans with 4 cm^−1^ resolution. For comparison, the infrared spectra were also recorded in transmission mode using KBr pellets with a Thermo Optics Avatar 330 instrument (Thermo Fischer Scientific, Waltham, MA, USA) in the 4000–400 cm^−1^ range, acquiring similar number scans and resolution.

Nitrogen sorption isotherms were recorded at 77 K using an ASAP 2010 Micromeritics apparatus (Micromeritics, Norcross, GA, USA). Each sample was degassed for several hours at 50 °C under secondary vacuum. The specific surface area, total pore volume, mean pore size and pore size distribution are determined from the N_2_ adsorption-desorption measurements.

XRD diffraction (XRD, Rigaku, Tokyo, Japan) was collected from samples pyrolyzed at 800 & 1100 °C using Cu Kα radiation at 40 kV and 30 mA. The powder diffraction pattern was recorded at 2*θ* = 10–80° with a 0.02°/s scanning rate.

The thermal evolution of SiOC samples was monitored by means of a thermogravimetric analysis (TGA) (Netzsch-Geratebau GmbH, Selb, Germany) with a Netzsch STA 409 thermobalance. The analysis was performed up to 1100 °C with a heating rate of 3.33 °C/min. To avoid any damage to the instrument, hydrogen could not be used as pyrolysis gas, so an inert argon flux of 50 cm^3^/min was employed. In this way it was possible to assess the ceramic yield of BTEM-derived SiOC without extracting the free carbon content.

UV-Vis-NIR spectra of bulk transparent ceramic aerogels were acquired with an Alignet Cary 5000 UV-Vis-NIR spectrophotometer (Alignet, Santa Clara, CA, USA) in the 400–800 nm wavelength range with a 600 nm/min acquisition rate. Transmittance data were normalized on a sample thickness of 3.1 mm, this resulting from the linear shrinkage of the two samples being different upon drying.

The morphological features of the SiOC aerogels were analyzed from fresh fracture surfaces using a Supra 40 Zeiss Field Emission Scanning Electron Microscope (FE-SEM) (JEOL JSM 6300F, Tokyo, Japan) after gold film deposition by sputtering.

## 3. Results

### 3.1. Density Measurements

The bulk density of the samples was measured from the volume and mass relationships. The average bulk density of the BTEM xerogel is about 0.75 g/cm^3^, which is slightly more than twice the density of the BTEM aerogel. The total porosity (*P*) by Vol. % can be determined from bulk and skeletal density measurements by applying the relation:
(1)
$$P=[1-\left(\rho_{b}/\rho_{s}\right)]\times 100\%$$

where *ρ_b_* is the bulk density and *ρ_s_* is the skeletal density.

The supercritical CO_2_ drying technique eliminates the capillary pressure generated at the liquid-vapor interface. This method of drying can maintain the integrity of the structure without any collapse. On the other hand, the atmospheric drying would induce the collapse of the network. The tendency of the collapse is inversely related to the size of the pores. The capillary pressure (*P_c_*) depends on the surface tension of the solvent (*γ*), the contact angle (*θ*), and pore size (*r*):
(2)
$$P_{c}=2\gamma \cos\theta /r$$


Both BTEM gels showed a linear shrinkage during the aging and drying processes. The linear shrinkage of both samples after complete gelation was found to be 6.95%, whereas the linear shrinkage after drying was about 31.81% for the ambient condition and 10.90% for supercritical carbon dioxide drying (see Table 2). The ambient dried samples usually result in dense and cracked materials, the so-called “xerogels”.

The BTEM and BTEMx dried gels have an average bulk density of 0.75 and 0.29 g/cm^3^, respectively. The average porosity BTEMx and BTEM is found to be 47% and 81%, respectively. The skeletal densities of the gels are also reported in Table 3. The skeletal densities of pre-ceramic networks are 1.36 and 1.53 g/cm^3^ for BTEMx, and BTEM, respectively.

At room temperature, both of the BTEM samples are transparent. The transparency of the samples at room temperature correlates well with the mean pore size determined via N_2_ physisorption. After the pyrolysis in H_2_ atmosphere at 800 °C, BTEMx samples are transparent and colorless, while the BTEM samples are less transparent and white in color. Furthermore, when the temperature is raised to 1100 °C, BTEMx samples become dark but transparency is maintained in some parts, whereas BTEM sample results tend to be brown and opaque (see Figure 4).

### 3.2. FT-IR

Infrared spectroscopy was used to detect the organic groups present in the starting molecular precursor (Figure 5). Some of these organic groups are eliminated at the later stages during the sol-gel and pyrolysis steps. The most strong bands at 1600 cm^−1^ and a shoulder at 1165 cm^−1^ are assigned to Si-O stretching in Si-OR groups. The bands at 2975 and 2880 cm^−1^ are assigned to the CH_3_ and CH_2_ groups [2,17]. In the spectra of the BTEM precursor, the peak around 1390 cm^−1^ was allotted to the stretching vibration of Si-CH_2_-Si bridges. The peak at 790 cm^−1^ belongs to asymmetric rocking vibrations of Si–CH_3_ and at 950 cm^−1^ these are related to Si–C stretching vibrations.

The infrared spectra of BTEMx xerogel and BTEM aerogel are shown in Figure 5. The peaks at 1160 and 1025 cm^−1^ are due to the formation of Si-O-Si bridges [17,18,19]^,^ and peaks at 805 and 675 cm^−^^1^ are assigned to the Si-O bending vibration. The peak around 805 cm^−1^ can also be allotted to Si-C vibration due to Si-C and Si-O overlapping in C-Si-O sites [17]. Another vibrational signal at 900 cm^−1^, assigned to the Si-OH functional group, forms during the hydrolysis reaction. This signal is an indication that Si-OH moieties remained unreacted during the condensation. The lower intensity peaks, detected at 2975, 1630 and 1362 cm^−^^1^_,_ may be assigned to C-H_x_, H_2_O and Si–CH_2_–Si bridges, respectively. Moreover, the presence of a broad peak around 3450 cm^−^^1^ belongs to the O-H vibration in Si-OH and adsorbed water. These peaks are also an indication for the formation of mesoporous materials, which is confirmed by N_2_-adsorption measurements having a high tendency to adsorb moisture from the atmosphere.

The synthesized BTEM and BTEMx samples showed similar peaks and intensities throughout the infrared spectra. These indicate that the ambient and supercritical drying do not have any effect on the formation of different functional groups. The formation of those functional groups depend only on the degree of both hydrolysis and condensation reactions.

The infrared spectra of BTEMx and BTEM specimens pyrolyzed in H_2_ atmosphere at 800 and 1100 °C are shown in Figure 6. At 800 °C, BTEMx and BTEM specimens present a small peak at 1278 cm^−^^1^ related to Si-CH_3_ formation. A strong overlapping band at 1185 and 1045 cm^−1^ belongs to the Si-O-Si stretching vibrations. The band at 800 cm^−1^ is usually assigned to Si-C and Si-O-Si vibrations. At 1100 °C, the band related to Si-O-Si and Si-C vibration shifts to the higher wavenumbers of 1050 and 820 cm^−^^1^_,_ respectively. Decarbonization at high temperatures eliminate Si-CH_3_ groups through the reaction proposed by Gobind Gas et al. to produce H_2_ and CH_4_ gases [17]. Therefore, the peak at 1278 cm^−^^1^ are not present at 1100 °C.

The infrared spectra measured in ATR show some differences from the ones obtained with the transmission mode. In ATR, the bands are distorted at a higher wavenumber and shift to lower frequencies [20]. The shift of peak positions to lower frequencies is challenging. When a functional group in the sample is identified by comparing a peak position in the transmission spectrum of a standard material to its accepted value, peak shifts due to ATR characteristics may be difficult to justify.

The FT-IR measurements in transmission mode at room temperature and 800 °C are revealed in Figure 7 for the BTEM aerogel samples. Some of the peaks which were not present in ATR mode are clearly visible in transmission mode. This is due to the fact that in transmission FT-IR spectra, the intensity is independent of the value of the wavelength. For a detailed explanation please refer to Section 4.

### 3.3. Nitrogen Adsorption-Desorption

The N_2_ physiosorption isotherms of all samples at 77 K are shown in Figure 8. The N_2_ isotherms obtained are all Type IV, which is the typical feature of the mesoporous materials. All BTEM aerogels pyrolyzed at different temperatures show H_1_ type hysteresis loops with well-defined cylindrical pore channels, whereas SiOC derived from BTEM ambigel shows a H_2_ type, indicating the pores are distorted due to the capillary stress. This can be justified by the fact that during the ambient pressure drying, BTEMx samples developed a higher capillary pressure, which finally reduced the size and deformed the cylindrical shape of the final pores.

Figure 9 shows the average pore size distribution of the samples treated at different temperatures. For all samples, the pores are in the range of 2–5 nm for BTEMx samples, and 2–20 nm for BTEM samples and their corresponding glass materials. The pore size distributions were evaluated through the Barrett–Joyner–Halenda (BJH) method from the desorption isotherms. Generally, the samples obtained from supercritical CO_2_ drying show wider pore size distributions coupled with slightly higher average pore size compared to the ones obtained from ambient pressure drying.

The specific surface area (SSA), total pore volume (TPV), and average pore size are shown in Table 4. The SSA reaches 1107 m^2^/g for BTEMx and 1140 m^2^/g for BTEM samples. At 800 °C, the SSA of BTEM aerogel is 890 m^2^/g with a TPV of 2.065 cm^3^/g, whereas for BTEMx xerogel the SSA is 760 m^2^/g with a TPV of 0.616 cm^3^/g. As the pyrolysis temperature is raised to 1100 °C, the SSA and TPV for both samples are reduced. The high SSA for BTEMx xerogel is attributed to the smaller pore size, whereas for BTEM aerogel the high SSA is derived from the high total pore volume.

### 3.4. XRD Analysis

Silicon oxycarbide is synthesized by incorporating C atoms within silica, giving a matrix in which Si atoms are bonded with both O and C atoms. The SiOC structure is based on a disordered network of Si-O and Si-C bonds. Based on previous studies, Si, C and SiC nanoparticles in the Si-O-C matrix are observed when the dried samples are pyrolyzed above 1000 °C [21]. Therefore, the XRD analysis are performed on the samples which are treated at 1100 °C.

Figure 10 shows the XRD profile of BTEMx and BTEM samples. At 1100 °C, the BTEM aerogel clearly reveals three main peeks at 28°, 47° and 56°, which reflect the formation of a cubic crystallite Si phase in the SiOC matrix [22]. The broad peak around 22° associates with amorphous silica. In the BTEMx sample at the same temperature, crystalline SiC and graphite phases can be formed through phase separation of amorphous SiOC into SiO_2_, SiC and C [23]. The presence of peaks at 26° and 43°, suggests the formation of turbostratic carbon or a graphene structure [24,25].

### 3.5. Thermogravimetric Analysis

Thermogravimetry was carried out on a BTEM aerogel to observe the ceramic yield and the thermal stability of the produced SiOC. Since no chemical differences between the xerogel and aerogel were spotted, this analysis is representative of both sample classes. The result of the analysis is given in Figure 11. An initial weight gain of 1.7% can be ascribed to the typical buoyancy effect of the thermobalance, which, summed to the weight loss under 150 °C, gives a total weight loss of −4.4% that can be assigned to adsorbed water. Such a result is in line with FT-IR data showing the presence of H_2_O signals. After the removal of water, the overall weight loss during polymer-to-ceramic transition under argon reaches −8.5% at 1100 °C, meaning that without extracting the free carbon by means of hydrogenation of -CH moieties, the ceramic yield of BTEM derived SiOC is equal to 91.5%.

The ceramic transition takes place starting at 200 °C with a small weight loss of −2.6%, followed by two endothermic steps of −4.2 and −1.7%, starting at 450 and 730 °C, respectively. The ceramization process involves first the volatilization of low molecular weight silanes, which slowly begins under 400 °C and finds a relevant acceleration over 450 °C, as suggested by the endothermic DTA signal. This process is normally ascribed to reorganization reactions between silyl groups, generating SiH_4_. Then, as the temperature surpasses 730 °C, molecular hydrogen and methane evolve, leaving a ceramic SiOC [26]. As expected, the pyrolyzed BTEM aerogel appeared blackish after the heat treatment, owing to the presence of a certain amount of free carbon in the ceramic network. Using hydrogen, the weight loss above 730 °C would clearly be more pronounced, as a bigger fraction of carbon would be extracted from the network. In any case, the ceramization process under argon flow can be considered representative of the described samples.

### 3.6. Transmittance Analysis

Ceramic BTEM and BTEMx gels treated at 800 °C in hydrogen became partially transparent to visible light. Therefore, UV-Vis-NIR spectra were acquired to properly define the transmittance of these two specimens over the visible wavelength range. As can be observed from Figure 12, the BTEM aerogel presents a higher transmittance than BTEM xerogel in the acquired λ range. Because of the Rayleigh scattering phenomena arising from the interaction with the huge amount of solid-gas interfaces at the pore walls, the UV-Vis spectrum of BTEM aerogel presents a steep absorbance characteristic that decreases with the photon wavelength [27]. On the other hand, BTEM xerogel shows a constant normalized transmittance of 0.2 over 520 nm, and then slowly decreases. Such a smaller transmittance can be explained considering that the bulk density of BTEM xerogel is higher than that of the relative aerogel, and because of its narrow porosity, the hydrogen decarbonization was less efficient, as confirmed by XRD spectra in Figure 12. Therefore, the overall transmittance of the SiOC xerogel is smaller than that of the aerogel, even if it is clear that light is preferably transmitted than scattered. It must be said that the analysed specimen presented a black dot in the irradiated area, and is most likely a rare carbon residue which is responsible of the overall low transmittance of BTEMx-800. However, comparing these data with the aspect of BTEMx-800 (Figure 4), it is clear that the actual transmittance of this ambigel is higher than the reported one.

### 3.7. SEM

The porous features of the SiOC samples were analyzed by means of FE-SEM acquisitions on fracture surfaces. The SEM images of preceramic and H_2_ treated BTEM aerogels are shown in Figure 13. The observations revealed the typical colloidal microstructure of sol-gel derived compounds with nanometric dimensions. The diameter of colloidal particles slightly increases after the ceramic transition as a consequence of densification phenomena. Such a result is confirmed by N_2_ physisorption analysis, which highlights a decrease in SSA and TPV. Finally, no inhomogeneities can be observed in the preceramic microstructure, suggesting that gelification occurs in a proper manner throughout the network synthesis.

## 4. Discussion

The peaks in infrared spectra in the ATR experiment are distorted at high wavenumbers, especially the ones which are pyrolyzed at high temperatures. In the ATR experiment, the depth of penetration *d_p_* is a function of wavelength (*λ*), incident angle of the beam (*θ*), the reflective indices of the ATR crystal (*n*_1_), and the sample (*n*_2_), and all are correlated by the following equation [21,28].

(3)
$$d_{p}=\lambda /2\pi n_{1}\sqrt{\sin\theta^{2}-{\left(n_{2}/n_{1}\right)}^{2}}$$


Assuming that the 
$n_{2}/n_{1}$
 ratio is constant, then the path of penetration is linearly related with the wavelength of IR radiation [21], and this results in the change in intensity in ATR. However, in transmission mode, the value *d_p_* is related to the thickness of the sample, which is constant.

For comparison, the FT-IR measurements in both ATR and transmission modes at room temperature and 800 °C are revealed in Figure 14. Some of the peaks in Figure 14a at higher wavenumbers are not visible. This is particularly true for the sample pyrolyzed at high temperatures (800 °C), and the relative intensity of the peaks at higher wavenumbers seems to disappear. However, when the transmission experiment is done for the BTEM sample at 800 °C (shown Figure 14b), the broad band appeared from 3200–3600 cm^−1^ which can be allotted to the H_2_O water or Si-OH groups. The peaks at 1280, 1470 and 1645 cm^−1^ belong to Si-CH_3_ vibrations, C-H bending and adsorbed moisture, respectively. Furthermore, the bands at 2980 cm^−1^ could be assigned to C-H vibrations in the –CH_3_ group and at 2855–2940 cm^−1^ are related to C-H vibration in the –CH_2_ group.

The transmission spectrum of the BTEM sample decarbonized at 800 °C shows intense peaks at 1060, 882 and 800 cm^−1^. In the ATR spectrum, these bands are shifted by 15, 0 and 0 cm^−1^ toward lower frequency, respectively. These large frequency shifts and peak intensity changes will reduce the performance of any computer search routine.

The BET equation is used to calculate the specific surface area (SSA) from 0.05 to 0.3 relative pressure. The high SSA of BTEMx and BTEM gels are 1107 and 1140 m^2^/g, respectively. The high surface area of the BTEM aerogel is due to its high TPV of 2.522 cm^3^/g. The presence of high total pore volume in the BTEM sample is associated with the use of supercritical CO_2_ drying, which maintained the integrity of the structural network. This type of drying method results in high porosity and TPV. However, high SSA of BTEMx xerogel comes with a smaller pore size. During the ambient drying, the integrity of the SiOC network shrinks too much and some parts of the xerogel sample showed cracks on the surface (see Figure 4) and possessed a smaller pore size. The average pore size of BTEMx xerogel is 3.4 nm, which is 2.6 fold smaller than that of the BTEM aerogel. In general, the surface areas (SA_BET_) are directly related to TPV or V and inversely related to the mean pore size according to Equation (4). As the pyrolysis temperatures increased to 800 and 1100 °C, the SSA and TPV of the samples decreased.

(4)
$$\mathrm{Average}\;\mathrm{pore}\;\mathrm{size}=4V/{\mathrm{SA}}_{\mathrm{BET}}$$


The BTEMx xerogel samples are more transparent than the BTEM aerogels. The transparency of the samples at room temperature correlate well with the mean pore size. This transparency can be seen up to 800 °C. However, the further increasing of the pyrolysis temperature to 1100 °C makes the BTEMx samples become dark, but the transparency is maintained in some parts, whereas BTEM samples become brown and opaque.

## 5. Conclusions

FT-IR and nitrogen sorption results revealed the successful synthesis of SiOC gels and the formation of mesoporous materials, respectively. From measurements, it is evident that the drying steps significantly affect the mean pore size and shape. Ambient dried samples resulted in smaller average pore sizes compared to the supercritical CO_2_ dried SiOC samples and therefore they appear totally transparent. It was also shown that the SiOC aerogels can be synthesized in crack-free monoliths and exhibit homogeneity and good transparency, being less than that of the xerogel. The synthesized SiOC samples present extremely porous structures, possessing 1140 m^2^/g SSA and 2.522 m^2^/g TPV. Generally, all the samples at room temperature show high surface areas (>1100 m^2^/g), that derive from a narrow distribution of small mesopores in the case of the xerogel, whereas those of the aerogel samples are associated with their high total pore volume. As pyrolysis temperature is raised to 800 and 1100 °C, the SSA and TPV for both samples are slightly reduced. The SSA of BTEM aerogels are 890 and 366 m^2^/g at 800 and 1100 °C, respectively. These SiOC ceramics have potential applications in photonics, gas sensors, and for Li-ion batteries, biosensors and photovoltaic cells.

## Figures and Tables

**Figure 1 materials-15-01277-f001:**
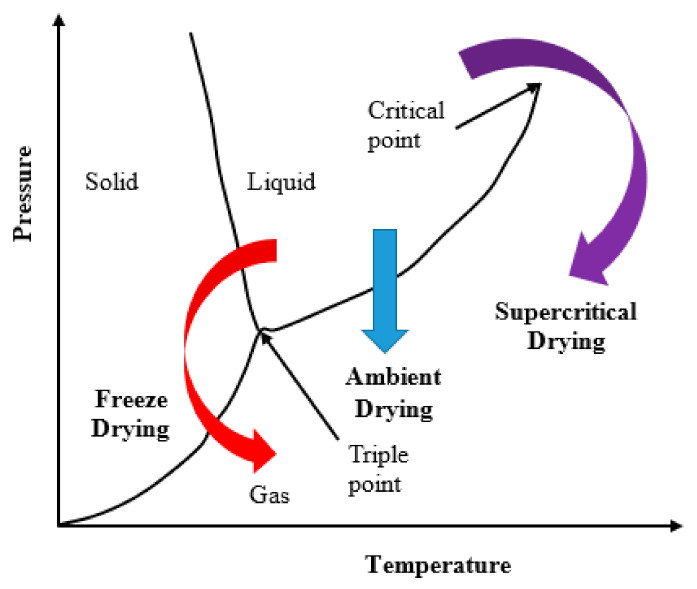
Schematic phase diagram of a pure compound with three basic gel drying routes.

**Figure 2 materials-15-01277-f002:**
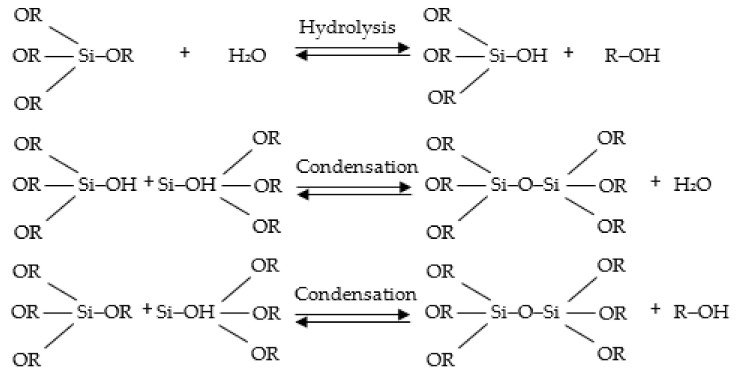
The hydrolysis—condensation reactions in the sol-gel process of alkoxysilane.

**Figure 3 materials-15-01277-f003:**
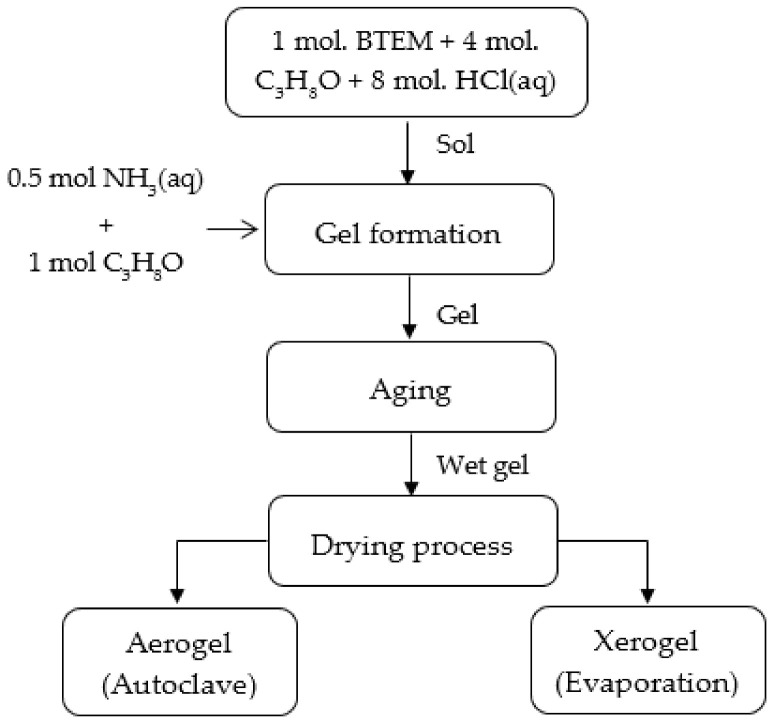
Flowchart for the preparation of BTEM gels.

**Figure 4 materials-15-01277-f004:**
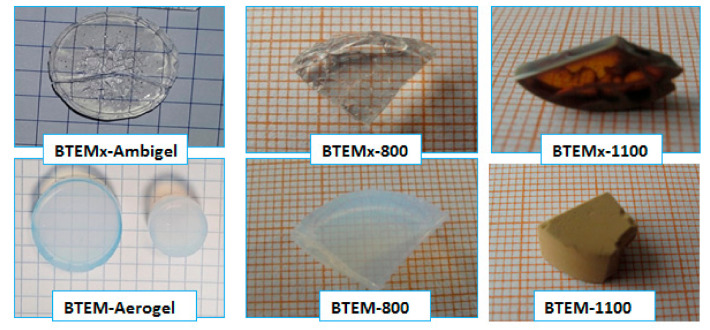
Photographs of the BTEMx-Ambigel and BTEM-Aerogel in pristine form, and heat treated at 800 and 1100 °C.

**Figure 5 materials-15-01277-f005:**
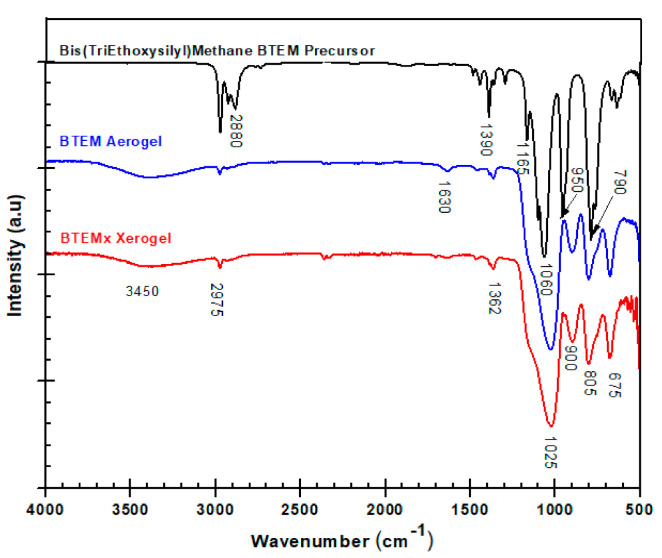
Infrared spectra of BTEM precursor, BTEMx xerogel and BTEM aerogel in ATR mode within 500–4000 cm^−1^ spectral range.

**Figure 6 materials-15-01277-f006:**
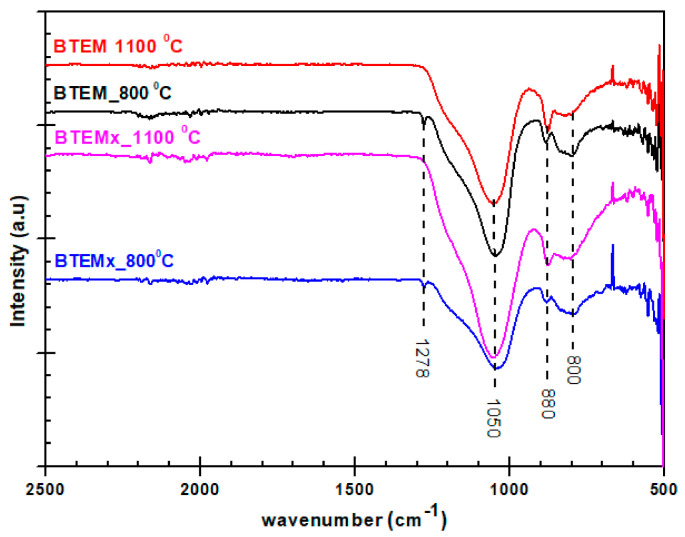
FT-IR spectra of BTEMx xerogel and BTEM aerogel samples pyrolyzed in H_2_ atmosphere at 800 and 1100 °C in the spectra range of 500–2500 cm^−1^.

**Figure 7 materials-15-01277-f007:**
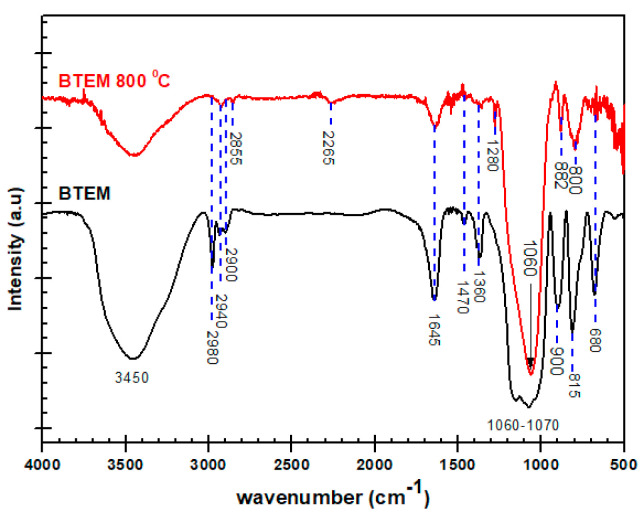
FT-IR spectra recorded in transmission modes for BTEM samples at room temperature RT and 800 °C within the 500–4000 cm^−1^ spectral range.

**Figure 8 materials-15-01277-f008:**
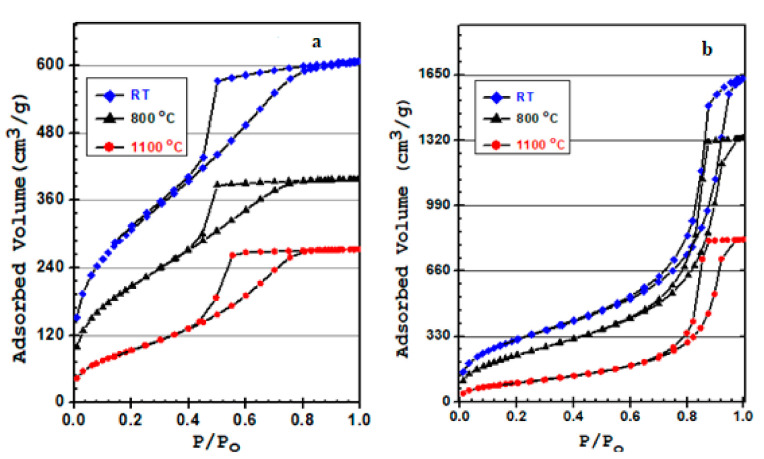
N_2_ adsorption-desorption isotherm of (**a**) BTEMx xerogel and (**b**) BTEM aerogel at different temperatures: room temperature RT, 800 and 1100 °C.

**Figure 9 materials-15-01277-f009:**
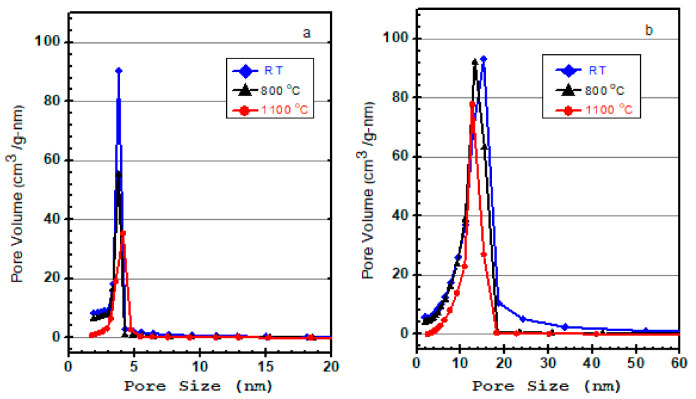
N_2_ adsorption-desorption isotherm of (**a**) BTEMx xerogel and (**b**) BTEM aerogel at different temperatures: room temperature RM, 800 and 1100 °C.

**Figure 10 materials-15-01277-f010:**
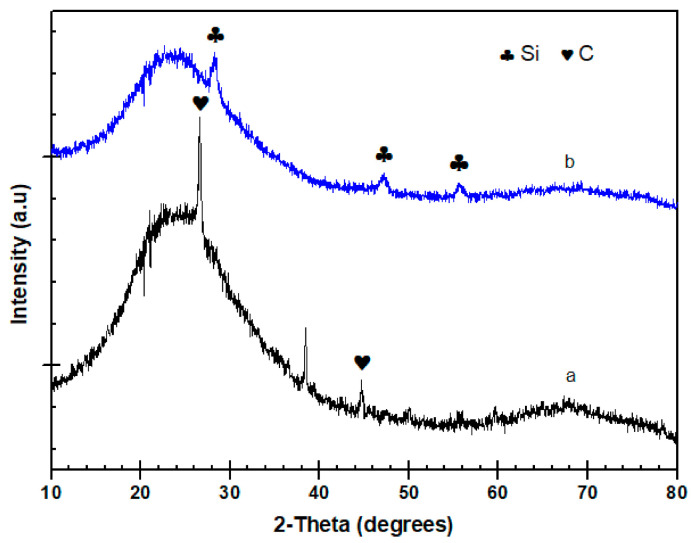
XRD patterns of (**a**) BTEMx xerogel and (**b**) BTEM aerogel SiOC ceramic samples decarbonized at 1100 °C in H_2_ atmosphere.

**Figure 11 materials-15-01277-f011:**
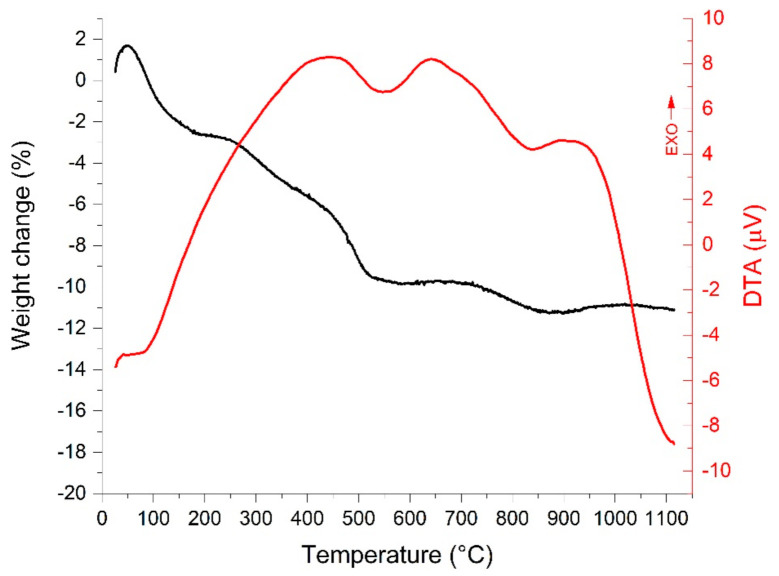
TG/DTA curves of BTEM aerogel pyrolyzed in argon up to 1100 °C at 3.33 °C/min. The TGA curve is reported in black, while the DTA signal is given in red.

**Figure 12 materials-15-01277-f012:**
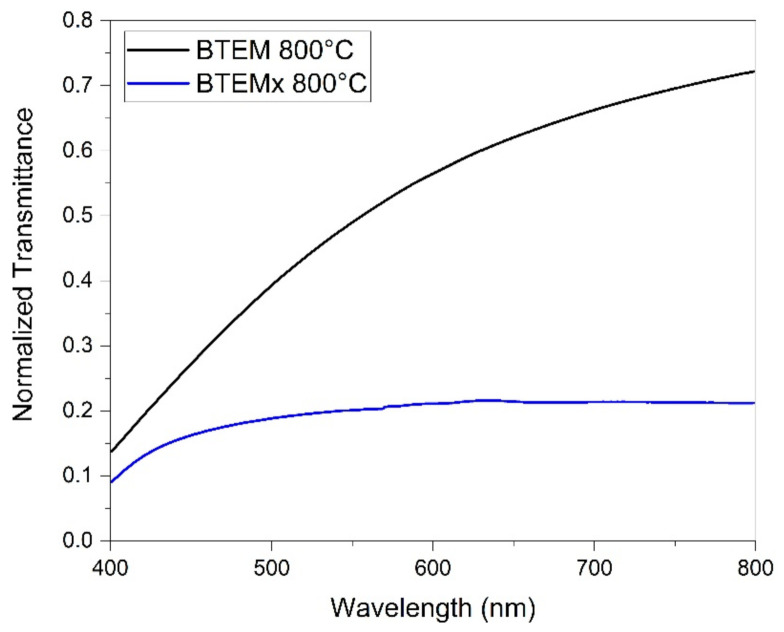
Normalized transmittance spectra of BTEMx and BTEM samples pyrolyzed at 800 °C.

**Figure 13 materials-15-01277-f013:**
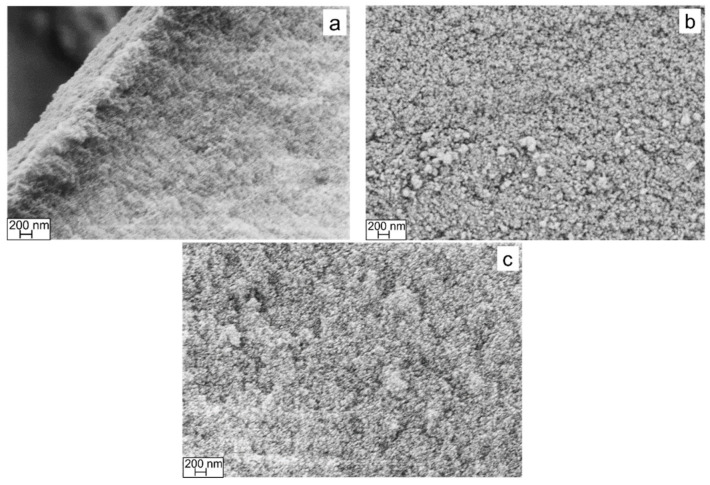
FE-SEM images of (**a**) BTEM aerogel, (**b**) BTEM aerogel at 800 °C and (**c**) BTEM aerogel at 1100 °C.

**Figure 14 materials-15-01277-f014:**
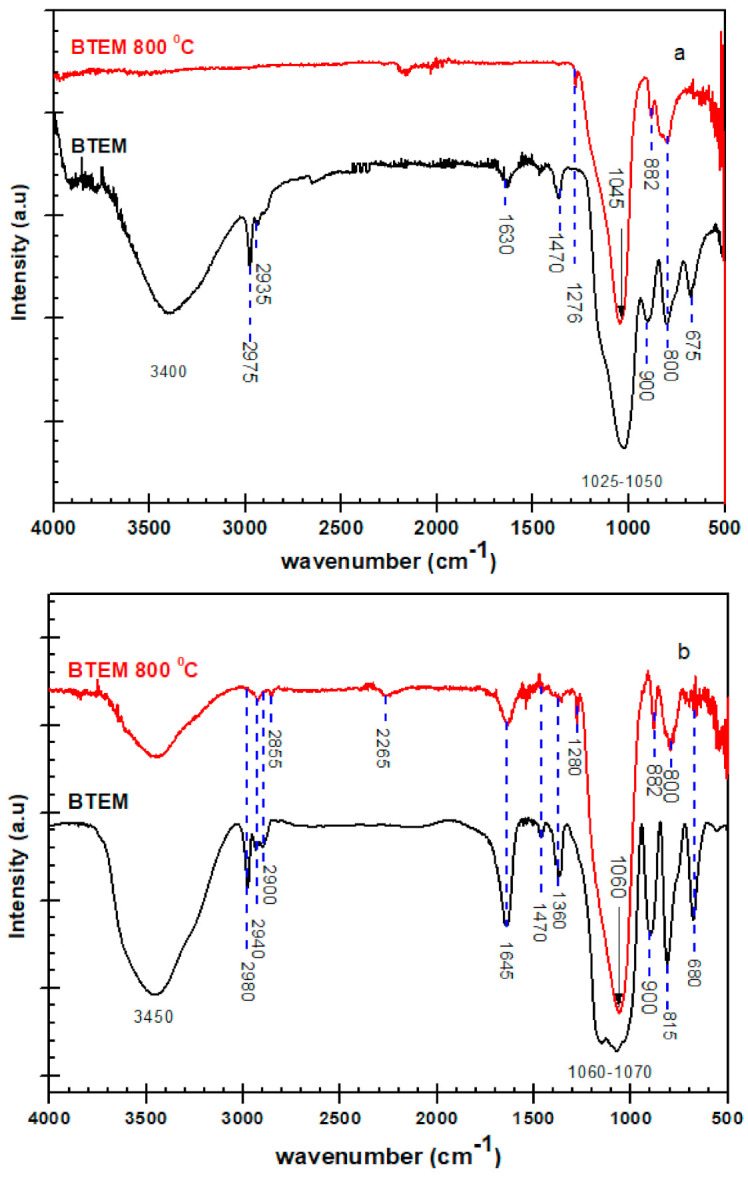
FT-IR spectra recorded in (**a**) ATR and (**b**) transmission modes for BTEM samples at room temperature (BTEM) and 800 °C (BTEM 800 °C) between 500–4000 cm^−1^ spectral range.

**Table 1 materials-15-01277-t001:** Sample labeling indicating drying methods.

Sample Label 1	Method of Drying	Remark
BTEMx	Ambient pressure drying	Xerogel/Ambigel
BTEM	Supercritical CO_2_ drying	Aerogel

**Table 2 materials-15-01277-t002:** Linear shrinkages of the wet and dried gels from BTEMx and BTEM samples.

Sample	Wet Gel (%)	Dried Gel (%)
BTEMx	6.95 ± 0.37	31.81 ± 1.89
BTEM	6.95 ± 0.37	10.90 ± 2.09

**Table 3 materials-15-01277-t003:** Bulk, skeletal density porosity of BTEMx and BTEM samples.

Sample	Bulk Density (g/cm^3^)	Skeletal Density (g/cm^3^)	Porosity (%)
BTEMx	0.75 ± 0.03	1.36	47
BTEM	0.29 ± 0.01	1.53	81

**Table 4 materials-15-01277-t004:** Specific surface area SSA_BET_, TPV and average pore size of SiOC samples.

Sample	SSA_BET_ (m^2^/g)	TPV (cm^3^/g)	Pore Size (nm)
BTEMx (RT)	1107 ± 9	0.940	3.4
BTEMx (800 °C)	760 ± 4	0.616	3.2
BTEMx (1100 °C)	358 ± 2	0.424	4.7
BTEM (RT)	1140 ± 8	2.522	8.8
BTEM (800 °C)	890 ± 3	2.065	9.3
BTEM (1100 °C)	366 ± 1	1.269	13.9

## Data Availability

Data available upon request.

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
