# Peer review of "Synthesis and Characterization of High Surface Area Transparent SiOC Aerogels from Hybrid Silicon Alkoxide: A Comparison between Ambient Pressure and Supercritical Drying"

_materials, 2022, doi:10.3390/ma15041277_

Round 1
Reviewer 1 Report
The authors describe the synthesis of highly porous and transparent silicon oxycarbide (SiOC) gels sized from Bis(Triethoxysilyl) Methane (BTEM) by the sol-gel technique followed by drying at ambient and supercritical pressure. This paper is well written, and the results presented are interesting and can be accepted for publication, after minor modifications:
* the stability of the synthesized gel should be discussed
* the scales must be added on the x-axis and the y-axis
* the transmittance axis should be added in figures 5, 6, 7, 8 and 12
Reviewer 2 Report
- What is the reason behind synthesising this type of material is not mentioned in the introduction, which is very imp to establish the necessity of this material
- Why is this method of synthesising unique or what is the novelty of this process. Establish a comparison table to set the benchmark of your work
- Correct the line spacing from line 123-127
- It is recommended to conduct thermal analysis like TG-DTA to understand the thermal stability and conversion temperature more accurately
- SEM images should be included to understand the porosity, morphology
- Also, transparency of the material should be noted
- Finally, application of these materials should be clearly mentioned
Reviewer 3 Report
In this paper, the authors synthesized the highly porous and transparent Silicon Oxycarbide (SiOC) gels using Bis(Triethoxysilyl) Methane (BTEM).
For this paper, the answer is “Major Revision”. Please include:
- At Introduction, authors must include the advantages of ambient pressure and supercritical CO2 drying methods employed to this study.
- Specify what does it mean “ambient pressure”
- At 3.2., in order to compare the samples, include the infrared spectra of BTEMx xerogel and BTEM aerogel in Figure 5 and eliminated the Figure 6.
- At 3.3., page 11, line 282, specify the full name for BJH method
- Page 11, line 289, modify “The SSA” with “The Specific surface area (SSA)”.
- At Conclusion, please, specify the applications of the prepared materials.

Round 2
Reviewer 2 Report
all comments are addressed
Reviewer 3 Report
The authors have improve the paper and it can be publish in this form.